# Comparative Hydrodynamic Analysis by Using Two−Dimensional Models and Application to a New Bridge

**Jesús Mateo-Lázaro [1],\*** , **Jorge Castillo-Mateo [2]** , **Alejandro García-Gil [3]**,
**José Ángel Sánchez-Navarro [1]**, **Víctor Fuertes-Rodríguez [4]** and **Vanesa Edo-Romero [1]**

[1]  Dept. of Earth Sciences, University of Zaragoza, Pedro Cerbuna, 12, 50009 Zaragoza, Spain;
    joseange@unizar.es (Á.A.S.-N.); vanesa_edo@hotmail.com (V.E.-R.)
[2]  Dept. of Statistical Methods, University of Zaragoza, Pedro Cerbuna, 12, 50009 Zaragoza, Spain;
    720193@unizar.es
[3]  Geological Survey of Spain (IGME), Ríos Rosas 23, 28003 Madrid, Spain; a.garcia@igme.es
[4]  Dept. of Geography and Territorial Planning, University of Zaragoza, Pedro Cerbuna, 12, 50009 Zaragoza,
    Spain; 696825@unizar.es
\*  Correspondence: jesmateo@unizar.es; Tel.: +34-(9)-690-609-948

**Abstract:** This document develops a methodology that evaluates the impact on the environment of the rivers produced by the creation of civil infrastructures. The methodology is based on the two-dimensional hydrodynamic calculation by using shallow water equations both in the conditions prior to the creation of the infrastructure, and in the new conditions after the infrastructure is created. Subsequently, several characteristics, such as water depth and velocity, among others, are compared between the initial and final conditions, and a two-dimensional zoning of the changes observed is obtained. The methodology herein presented is useful to verify the impact that the implantation of different infrastructures around the river currents could produce. In addition, it is also relevant for carrying out a study taking into account different infrastructure options related to river currents, as well as for selecting the most suitable one. By using the methodology presented, changes on the regime of the currents caused by the infrastructures can be deduced, including a qualitative and quantitative zoning of the changes, with a special emphasis on some characteristics, such as depth and velocity. The methodology is applied in a case study for the creation of a road bridge over the Jalon River in Spain.

**Keywords:** shallow water equations; floodway; floodplain; computational Fluid Dynamics (CFD); SHEE software; Iber program

---

## 1. Introduction

The public institutions of countries and communities are responsible for regulating the actions that are carried out in the hydraulic domain of rivers and water bodies. When new infrastructures are created in rivers, such as bridges or water diversion spillways, there are three main concerns that need to be controlled: (1) the increase or generation of risks for the environment around the created infrastructure, for example, the ones derived from overpopulated centers, (2) the security for the users of the infrastructure to be created, and (3) the conservation of the functions and natural conditions of the river itself. Normally, the state agencies impose the control and evaluation criteria for the modifications derived from the new infrastructures.

This work aims to apply a control methodology for the three aforementioned aspects. The methodology is based on the comparative evaluation between the initial state of the river

and the state once the infrastructure is created. To carry out this study, two specific technologies are used. The first one is a two-dimensional numerical model that is able to simulate the flow of water and its main characteristics, such as flow, water velocity, and water depth. The model is generated using a free software called Iber, which was developed by State Agencies and Spanish Universities. The second technique is based on a software able to interpret and compare the results obtained from the two-dimensional numerical model. This model is called SHEE (Simulation of Hydrological Extreme Events), and was developed by the Authors of this paper in the Department of Earth Sciences at The University of Zaragoza. This model has given rise to numerous professional works, research projects and scientific publications.

Other hydrodynamic modeling approaches are implemented by employing 1D, 2D or 1D/2D schematizations. Although one-dimensional hydrodynamic models are still in widespread use for many applications [1–8], the use of two-dimensional models is required in urban areas to reproduce the complex, multidirectional flow paths generated by urban features [9–12].

Most of the hydraulic studies on infrastructures or next to rivers are focused on assessing the flooding risk of the infrastructure itself or on the protection of the facilities attached to rivers [13]. It should be noted that, in addition, this study tries to assess the possible increase in flooding that new infrastructures can generate on areas annexed to rivers, including urbanized areas.

In relation to the creation of infrastructures and their influence on river currents, there are studies that have quantified the risk of flooding, both considering or disregarding the engineering protection measures, and using the hydraulic approach [14]. The existence of river defense structures is restricted by several factors, such as the unavailability of space, the geotechnical limitations, and aesthetic reasons [15]. Studies have found that individual protection measures (IPM) have the potential to mitigate flood risk [16].

## 2. Materials and Methods

### 2.1. Hydrodinamic Model

The numerical model used in this work is the Spanish computer program called Iber, which is a two-dimensional model for the simulation of free surface flow in rivers and estuaries. It is distributed for free in English and Spanish (http://www.iberaula.es/). The Iber application has been generated to obtain a hydroinformatic Computational Fluid Dynamics (CFD) tool for free surface flows. Various academic disciplines such as hydraulics, hydrology, fluid dynamics, soil dynamics, chemistry, etc. are involved. The multidisciplinary creator team comes from different public institutions and universities, mainly located in Spain, such as the Group of Engineering of the Water and the Environment of the University of *La Coruña*, the group of Fluvial Dynamics and Hydrological Engineering of the Flumen Institute and the International Center for Numerical Methods in Engineering (both from the Polytechnic University of Catalonia), and the Spanish Center for Hydrographic Studies (CEDEX), as well as researchers from other universities like the University of Santiago de Compostela, the University of Vigo, and the Institute for Environmental Sciences of the University of Genoa.

Iber solves the full depth-averaged shallow water equations or two-dimensional equations of St. Venant (Equations (1)–(3) in order to compute the water depth and the two horizontal components of the depth-averaged velocity. These equations assume a hydrostatic pressure distribution and a relatively uniform distribution of the velocity with the depth. The equations are solved with an unstructured finite volume solver explicit in time.

$$\frac{\partial h}{\partial t} + \frac{\partial h U_x}{\partial x} + \frac{\partial h U_y}{\partial y} = M_S \tag{1}$$

$$\frac{\partial h U_x}{\partial t} + \frac{\partial h U_x^2}{\partial x} + \frac{\partial h U_x U_y}{\partial y} = -gh\frac{\partial Z_s}{\partial x} + \frac{\tau_{s,x}}{\rho} - \frac{\tau_{b,x}}{\rho} - \frac{g}{\rho}\frac{h^2}{2}\frac{\partial \rho}{\partial x} + 2\Omega \sin\lambda U_y + \frac{\partial h \tau_{xx}^e}{\partial x} + \frac{\partial h \tau_{xy}^e}{\partial y} + M_x \tag{2}$$

$$\frac{\partial hU_y}{\partial t} + \frac{\partial hU_xU_y}{\partial x} + \frac{\partial hU_y^2}{\partial y} = -gh\frac{\partial Z_s}{\partial y} + \frac{\tau_{s,y}}{\rho} - \frac{\tau_{b,y}}{\rho} - \frac{g}{\rho}\frac{h^2}{2}\frac{\partial \rho}{\partial y} + 2\Omega \, \sin\lambda U_x + \frac{\partial h\tau_{xy}^e}{\partial x} + \frac{\partial h\tau_{yy}^e}{\partial y} + M_y \qquad (3)$$

where h is the water depth, $U_x$, $U_y$ are the horizontal velocities averaged with depth, g is the acceleration of gravity, $Z_s$ is the elevation of the free surface, $\tau_s$ is the stress on the free surface due to friction caused by the wind, $\tau_b$ is the stress due to the friction of the bottom, $\rho$ is the density of water, $\Omega$ is the angular velocity of rotation of the earth, $\lambda$ is the latitude of the point considered, $\tau^e_{xx}$, $\tau^e_{xy}$, $\tau^e_{yy}$ are the horizontal effective tangential stresses, and $M_s$, $M_x$, $M_y$ are the source/sink terms of mass and of momentum, respectively. The source terms in the hydrodynamic equations include hydrostatic pressure, bottom slope, viscous and turbulent tangential stresses, bottom friction, surface wind friction, precipitation, and infiltration.

By using Iber, a large number of numerical models can be made with applications in hydraulics and river morphology. The calculation of flow in rivers, the definition of flood zones, the evaluation of risk areas, and the delimitation of roads of intense drainage, are all assumptions where the flow is two-dimensional, and are some of its fundamental applications. It includes the sediment transport and the bottom transport, along with the tidal flow in estuaries, among other possibilities. The simulation of water passage under bridges, sluices, and spillways can also be done including the effect of wind, and it is possible to model the breakage of rafts and dams.

The Iber program simulates non-permanent flow or permanent flow, while maintaining a constant flow. The simulations need, firstly, a digital terrain model (DTM) that can generate two types of mesh: structured (e.g., a GRID model) and unstructured (e.g., a TIN model). Another necessary feature is the roughness of the terrain, which can be incorporated through a GRID coverage. Finally, it is fundamental to establish the boundary conditions of the model which, in an upstream case, consist of the hydrograph entering the domain of calculation. The downstream boundary condition refers to the regime in that section, which may be a critical, supercritical or sub-critical regime. As a result, the Iber program provides different coverages in GRID format. The most used modules are the water depth coverage and the total velocity ones.

The main information and reference sources for Iber main modules are described in [17]. There are also specific references for other modules, such as Numerical Schemes and Hydrological processes [18], Water Quality [19], and IberPLUS module [20]. Other publications investigate specific topics, such as the analysis of flooding scenarios in general [21], flood forecasting using coupled hydrological and hydraulic Models [22], extreme flood inundation in coastal river reaches [23], sensitivity of flood loss estimates for building representation and flow depth attribution methods in micro−scale flood modeling [24], real-time prediction of flood inundation [25], numerical distributed modeling of sedimentary and erosion processes [26], soil erosion predictions [27], bedload transport for mixed flows [28], sediment transport during the breakage of dams [29], drying and transport processes in distributed hydrological modeling [30], wood transport and sediment in rivers, flash floods, etc. [31–36], hydrodynamic movement in water bodies due to wind drag action [37], inundation modeling in coastal urban areas [38], local rainfall dynamics and uncertain boundary conditions [39,40], conservation of the cultural heritage due to flood risk [41], historical reconstruction [25], hazard mapping [42], research on the rivers confluence in order to analyze the geomorphic consequences of the backwater effect [43], assessment of the physical habitat suitability for fish [44], and many other works with a wide variety of themes.

## 2.2. Comparative Analysis

Once the coverage of water depth (h) and velocity (v) in GRID format have been obtained with the Iber program, a third coverage is acquired as the product h·v in each mesh node, through the SHEE program. The unit of this product is $m^2$/s and represents the specific stream−flow through a 1 m wide vertical strip perpendicular to the velocity vector. This feature is a measure of the Flood Hazard Rating (FHR) [45–47] where zones showing values h > 1, v > 1, or h·v > 0.5 are normally considered dangerous flood zones.

In essence, the comparative analysis consists in establishing the differences at each point of the calculation domain between the initial state (without bridge) and the final state (with bridge), with the coverage obtained, h, v, and h·v. Once these differences are obtained, it is a matter of observing and carrying out a discussion on the characteristics observed in each area of the calculation domain.

Floodway zone is understood as the area through which the avenue of 100 years of return period would pass without producing a water level over-elevation greater than 0.3 m, with respect to the entire existing floodplain. The over-elevation may be reduced to 0.1 m when the increase in flooding could lead to serious damage, or it could otherwise be increased up to 0.5 m in localized areas or when the increase in flooding would cause reduced damage. In the case of bridges, the State Administrations force not to invade the floodway zone with the abutments of the bridge, and to verify this, it is necessary to carry out simulations with and without the planned infrastructure. These criteria can be found in the Spanish legislation, which is shared by numerous European, American and United Kingdom institutions [45–47].

### 2.3. Description of the SHEE Software

Regarding the SHEE program, it derived from several publications, including those related to hydrology [48–59]. This program uses powerful libraries (e.g., OpenGL, GDI, GDAL, Proj4) for the management and display of DEM and datasets. Additionally, its interface provides rapid and high quality OPENGL graphics, in both RASTER and VECTOR formats. In this work, we have used the characteristics of the program that are capable of processing the results in GRID coverage format derived from the Iber program, which mainly consists of a map algebra to perform different comparatives.

GDI (Graphics Device Interface) is the Microsoft Windows application programming interface and core operating system component responsible for representing graphical objects and transmitting them to output devices, such as monitors and printers. OPENGL (Open Graphics Library) is the computer industry's standard application program interface (API) for defining 2D and 3D graphic images. GDI is very powerful in displaying raster images, while OpenGL has better performance with vector operations, and has the ability to generate 3D graphics and stereographic vision. GDAL (Geospatial Data Abstraction Library) is a translator library for raster and vector geospatial data formats that is released under an X/MIT style Open Source license by the Open Source Geospatial Foundation. As a library, it presents a single raster abstract data model and vector abstract data model to the calling application for all supported formats. PROJ4 is a library for performing conversions between cartographic projections. The library is now an OSGeo project.

The software SHEE has numerous applications for either DEM management or hydrological processes simulation. Obtaining new cartographic coverage with the combination of DEM and simulated processes is also possible. The DEM management is achieved using the GDAL, which permits the import and export of different archive formats, and can create new coverage from multiple archives. The program can combine coverage with different coordinate systems, thanks to the use of the PROJ4 library from the USGS. Thousands of terrestrial geodetic systems can be represented, transformed, and converted between them. To do that, the program is able to obtain necessary Spatial Reference Organization parameters from the internet server transfer. Downloading information from the WMS remote server is also possible. With regard to DEM characteristics, SHEE program can manage any format, size, accuracy, and reference system. E.g., Global DEM has been used like SRTM30 (with file size 3.6 Gb and grid size 30″, about 900 m), MDT5 of Spanish territory (120 Gb and 5 m), Lidar, etc. The use of Parallel Linear Reservoir (PLR) models as a hydrological model integrated within the sequential processing algorithm of the catchment is a special case of hydrological application, where every cell of the DEM is considered as a reservoir combination in parallel.

Future developments of SHEE program are very promising. The process of generating new textures can be performed by a Graphics Processing Unit (GPU), thereby making real-time processing very effective, and providing the possibility of displaying the simulation of geological structures in motion. Regarding the Graphics Processing Units, and since the DEM is becoming denser, we are currently

completing the development of hydrological models with this technique through the sequential processing algorithm, whose main advantage is the shortening of the computation time, which can be reduced 100 times. Due to parallel processing, it is necessary to reprogram the sequential algorithms for computing drainage networks.

## 3. Study Case for a New Bridge

The exposed methodology has been applied to a case of implantation of a new ring road in Sabiñan (Spain), which crosses the Jalon River downstream of the urban area. For this purpose, a bridge of 109 m in length has been designed, with sufficient hydraulic capacity for a flow greater than 500 years of return period. In Sabiñan, the flows of the Jalon River for return periods of 10, 100, and 500 are 199, 371, and 535 $m^3/s$, respectively. Figure 1 shows the valley with the implementation of the road infrastructure and a longitudinal section, along with the water level for different return periods. This bridge has a length of 109 m, and is divided into four spans: two 26 m central spans limited by the piers of the bridge, in which the main channel is located; and another two spans at the ends. Both measure about 24 m and are limited by the abutments of the bridge. The piers consist of three pairs, with a transverse separation of 4 m, with a foundation based on piles. They will have rounded edges, thus determining circular sections to improve the hydraulic behavior.

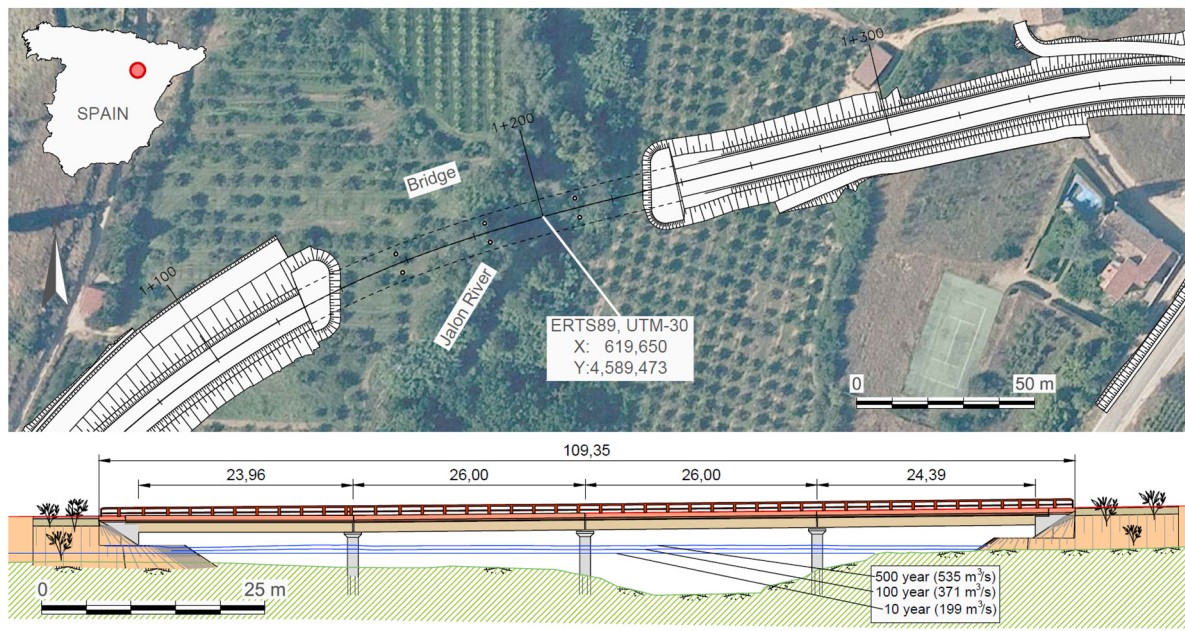

**Figure 1.** Road crossing on the Jalón River in Sabiñán (Spain), with a four-span bridge.

To carry out the simulations with the Iber program, a digital terrain model (DTM) has been used in a GRID format of 2 m mesh size, and obtained from a high-resolution LIDAR flight. To visualize the DTM in the different figures, coverage has been obtained by applying the algorithm of Sky-view Factor (SVF), a shading technique that has abundant literature references [60–62]. It allows us to perfectly distinguish the different morphologies of the terrain. Since the computational load of the SVF algorithm is very high, it has been programmed using parallel computing units (GPU) within the SHEE software.

The calculation domain is a 2180 × 730 m rectangle. Since the DTM is a GRID model of 2 m size, the total number of elements in the initial structured mesh is 397,850. These elements are converted into triangles to generate a Triangulated Irregular Network (TIN) mesh with twice as many elements: 795,700. Subsequently, an unstructured TIN mesh is generated, reducing the number of elements according to the criterion of 0.5 m tolerance, 15 m maximum side, and 1 m minimum side for triangular elements. With these criteria, the calculation domain has been reduced to 167,479 triangular elements

in an unstructured mesh. The piers of the bridge are made up of triangular elements with altimetry similar to that of the deck of the bridge, which is why they are represented in the DEM and, therefore, are taken into account within the hydrodynamic calculation.

Considering an axis centered on the flood plain, the river stretch of the calculation domain has a length of 2.5 km. If we measure the main channel, which has meander morphology in some parts, the river length increases to 3.3 km. The bridge is located away 740 m from the downstream edge of the calculation domain, and 1,450 m away from the upstream edge of the calculation domain. The boundary conditions of the model are located at these edges of the calculation domain.

The 100-year return period model has been calibrated with the Manning roughness coefficient, by comparing the water level deduced from the National System of Flood Zone Cartography (SNCZI) [63] of the Spanish Ministry of Environment, which was obtained from the simulation, resulting in n = 0.05 for the general case and 0.1 for the boundary condition in the downstream edge. The differences obtained in the calibration are: a bias of 24 cm and a deviation of 23 cm for the entire calculation domain. The Iber model is a non-permanent flow model. The calculations made in our research have been carried out by introducing a theoretical hydrograph that rises from 0 to the peak flow (371 m$^3$/s in the case of a 100-year return period), and remains constant from there for long enough so that there are no changes throughout the calculation domain. Thus, the simulations can be considered as a permanent flow corresponding to the peak flow.

## 4. Results and Discussion

This section is divided into two subsections: the first tries to select the best among the three geometric bridge options. The second subsection addresses the option selected as the most suitable one, and a series of more in-depth studies are carried out, comparing the simulations between the initial state (without bridge) and the final state (with bridge).

### 4.1. Study of Bridge Alternatives and Selection of the Most Suitable One

To reach the ideal solution, four cases have been studied: river without bridge, and river with three bridge options:

(1) Option 1: bridge with three 26 m spans and two pairs of piers.

(2) Option 2: same configuration as that for Option 1, plus a 10 m culvert hydraulic.

(3) Option 3: same configuration as that for Option 1, but adding another 26 m span and a pair of piers, equating 4 spans in total. The option represented in Figure 1 corresponds with option 3.

To obtain the surcharge (increase of flood elevation) produced by the bridge implantation, it is necessary to perform a simulation without a bridge and another one with a bridge. Figure 2 shows the water depth results for both option 3 situations.

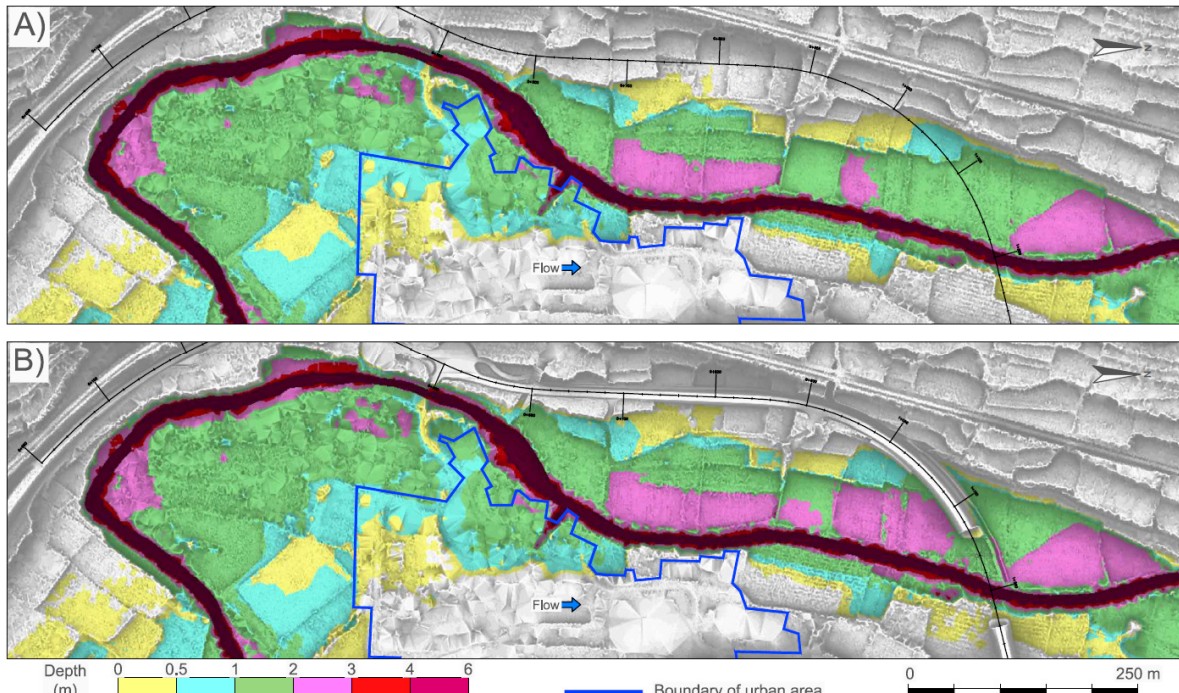

**Figure 2.** Water depth coverage: (**A**) initial state without bridge and (**B**) final state with the bridge from option 3. In this Figure and the following one, for the sake of detail, only the most representative part of the calculation domain is represented.

Comparing the water depth results, Figure 3 shows the increase of flood elevation occurred in each bridge option compared to the original setting. Several features stand out:

(1) Options 1 and 2 show a surcharge of 20 cm in the urban area, thus not meeting the established criteria of not exceeding a 10 cm surcharge in urban areas.

(2) Options 2 and 3 show a surcharge excess of 30 cm for a large area of the river, thus not meeting the criteria of not exceeding an increase of 30 cm of flood elevation and, therefore, affecting the floodway.

(3) Option 1 presents 80 cm local increase of flood elevation, being 60 cm in the case of option 2. In both cases, the criterion of not exceeding 50 cm in special cases is not met.

(4) There is a decrease in water depth downstream.

(5) Option 3 meets all the criteria of Section 2.2, so it turns out to be the most suitable option.

The hydraulic study shows that option 3 is the only option meeting the criteria established in Section 2.2 and, therefore, it is the case that will be developed more extensively in this article.

These examples have illustrated the great usefulness of the tools presented, both the numerical model Iber and SHEE that performs different geospatial algebra operations with Iber's results. In addition to the example presented, simulations can be carried out on other types of infrastructure, playing with different designs and locations, such as levees arranged on the banks of the rivers, limiting the floodplain, weirs for water intakes in channels, hydraulic waterfalls systems in steep creeks, floodgates in tidal flats, and many others.

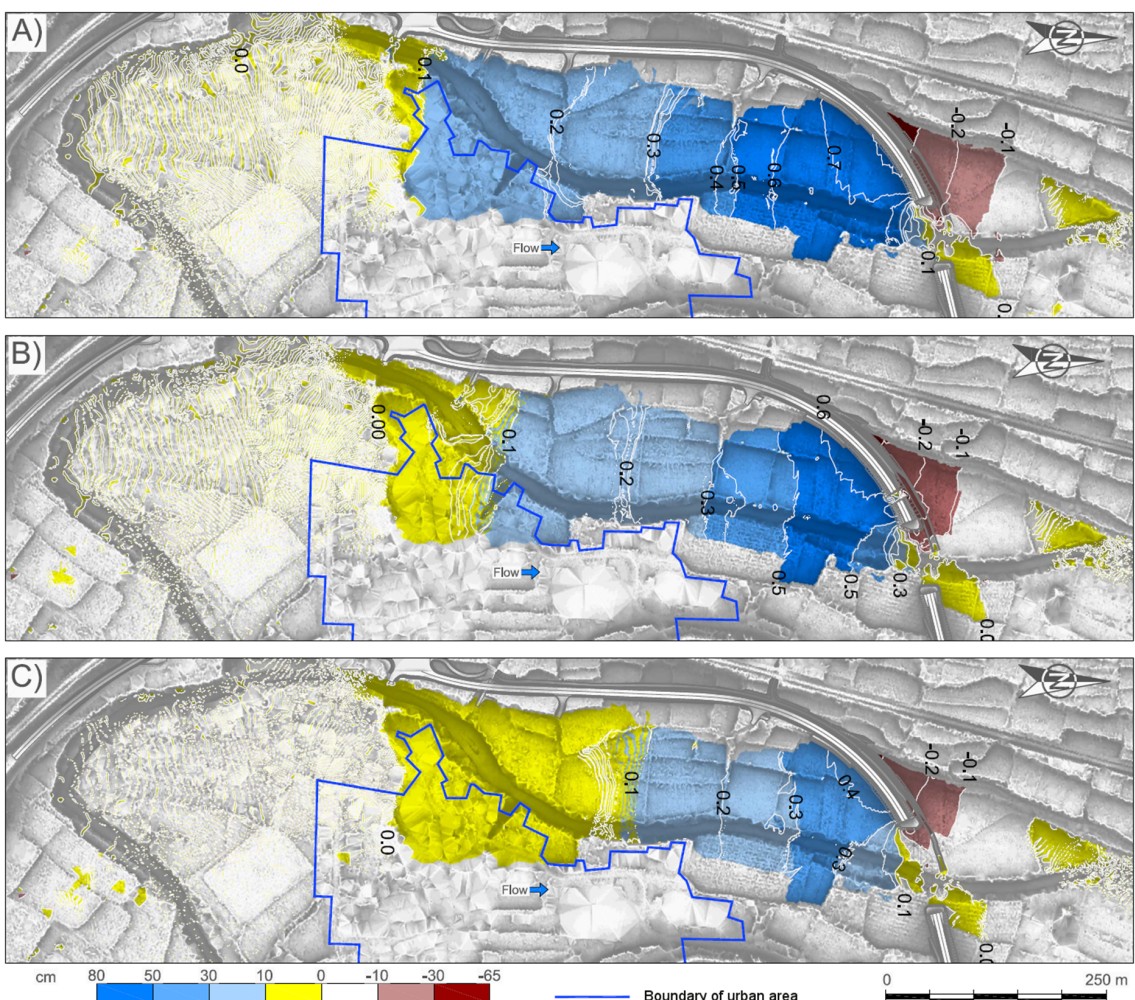

**Figure 3.** Comparative analysis of the water depth for the three options studied: (**A**) Option 1; (**B**) Option 2, and (**C**) Option 3. All of them show the variation in the water level produced by the execution of each option.

### 4.2. Hydraulic Detail Study of the Suitable Alternative (Option 3)

In addition to the water depth (Figure 2), the velocity of the flow has been studied. Figure 4A,B shows the spatial distribution of the velocity for the initial state (without bridge) and the final state, with option's 3 bridge. Figure 4C shows the difference between both coverages. The observations that can be made from these results are the following:

(1) In the flooded urban area, there is a large area where the velocity does not exceed 1.0 m/s, but there is another part where velocity exceeds 2.0 m/s, both in the initial and the final state.

(2) However, there is no significant increase in the velocity of the current due to the implantation of a bridge in the flooded urban area.

(3) Around the area where the abutments of the bridge are located, and in the strip corresponding to the floodway, there is a velocity increase of about 1.0 m/s, which becomes 1.9 m/s in the main channel (pink, light red and dark red colors).

(4) In the floodplain of the left bank of the river, there is a decrease in velocity, both upstream and downstream of the left abutment. This decrease in velocity is more significant downstream, where it decreases by −1.4 m/s compared to the velocity variation observed in the upstream area, whose maximum variation is in the order of −0.5 m/s.

(5) In the main channel area, there is a decrease in draft (yellow color in Figure 4C) that reaches more than 300 m upstream, which is longer than the decrease in the floodplain.

(6) The law of conservation of mass explains the inverse relationship between water depth and low velocity. For example, in the section of the bridge and in the adjacent area downstream, there is an increase of velocity, coupled with a decrease in water depth. In contrast, in the upstream floodplain zone, where the velocity decreases, an increase in water depth is observed.

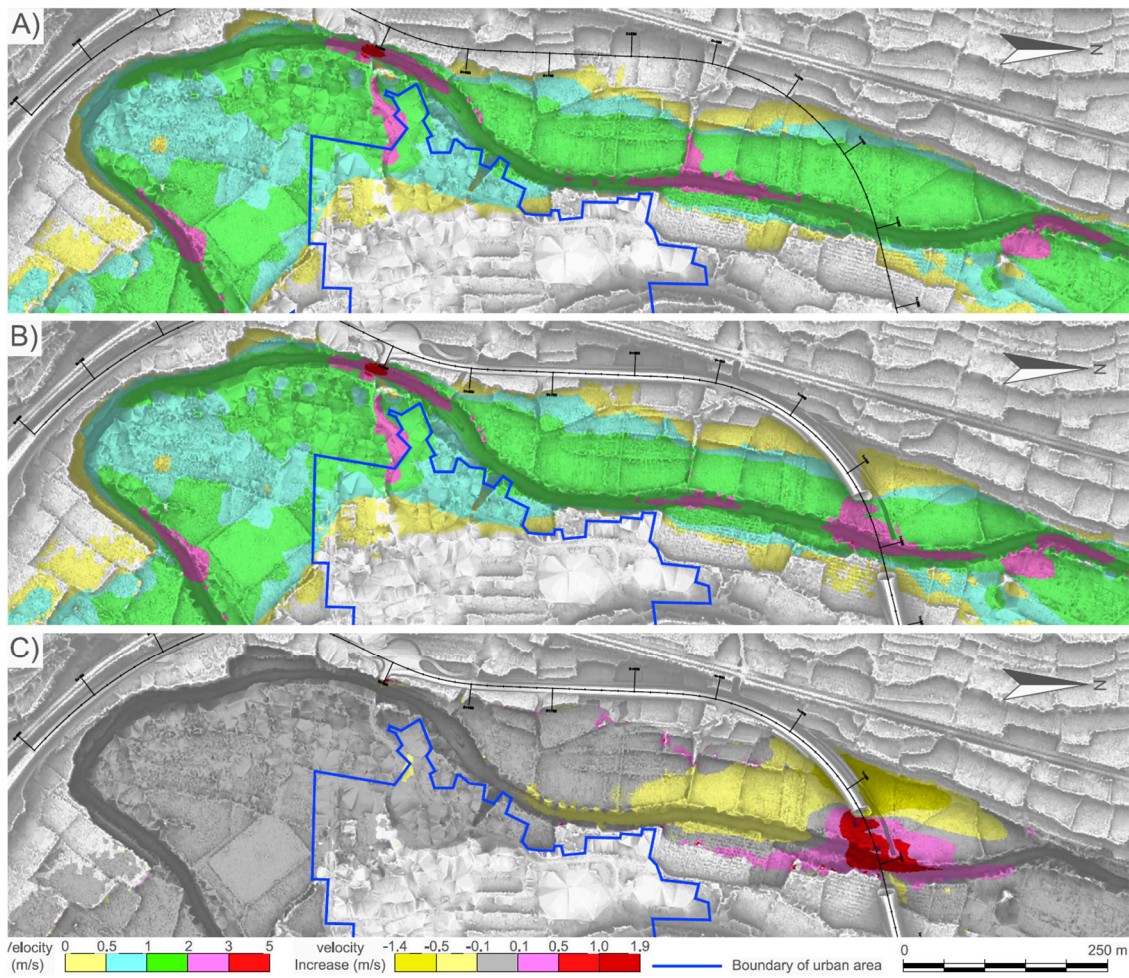

**Figure 4.** Velocity coverage: (**A**) initial state without bridge and (**B**) state with the option bridge 3. In (**C**), the difference between the two previous velocity coverages is shown.

Finally, we have studied the h·v relationship or Flood Hazard Rating (FHR) shown in Figure 5. In A and B, results show the initial state (without bridge) and the final state (with bridge), respectively. C shows the increase between both coverages. The observations can be synthesized in the following points:

(1) If we compare Figure 3C, Figure 4C, Figure 5C, it follows that, for the coverage h·v, the velocity term (v) has a greater influence than the depth term (h).

(2) The previous feature is also explained when analyzing the differences of depth, which are within the order of 0.3, 0.4 m (see Figure 3), compared to the variations in velocity (see Figure 4), which are greater, within the order of 1.0 m up to 1.9 m.

(3) As with velocity, there is a significant increase in FHR that reaches 4.0 m²/s in the main channel, and 2.5 m²/s in the rest of the floodway around the bridge section.

(4) The decrease in FHR is more significant downstream of the bridge, reaching −2.5 m²/s compared to the upstream area, which only reaches −0.5 m²/s.

(5) In the main channel area, there is a decrease in FHR (yellow in Figure 5C) that extends more than 500 m upstream, which is longer than the decrease in floodplain, which barely reaches 100 m.

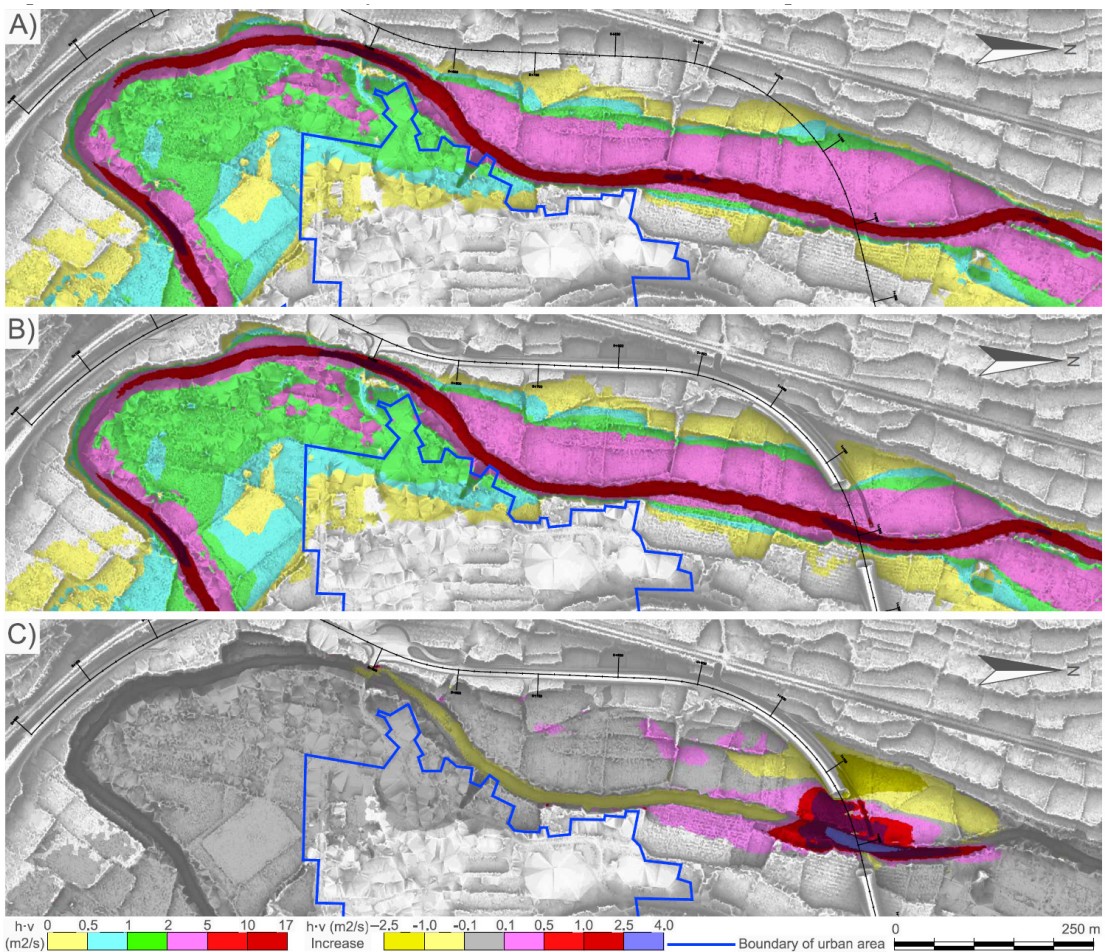

**Figure 5.** Coverage Flood Hazard Rating (FHR) (h·v) for (**A**) initial state without bridge, and (**B**) state with option's 3 bridge, (**C**) representation of the difference between the two previous coverages.

## 5. Conclusions

In this article, a two-dimensional hydrodynamic calculation methodology has been presented through the use of the Spanish computer program Iber, which is freely distributed. It was applied to a case study for the implementation of a road bridge over the Jalon River in Sabiñan (Spain). With the results of the hydrodynamic study, water depth and velocity coverage values were generated for the initial state without a bridge, and for the final state with a bridge. Using the SHEE program of the University of Zaragoza, FHR coverage was generated. In addition, a comparative study was carried out with the variations between both scenarios: initial and final. The most relevant conclusions are described below:

(1) The methodology presented is useful to verify the impact the implementation of different infrastructures may have on river flows.

(2) The methodology presented is useful for carrying out a study with different infrastructure options related to river currents as well as for selecting the most suitable one.

(3) By using the methodology presented, changes generated on the currents regime by the infrastructures can be deduced, including a qualitative and quantitative zoning of the changes produced, especially for features such as depth, velocity and FHR.

(4) Based on the case study presented, various events, such as an increase in water depth in the upstream area of the bridge and a decrease in the downstream area can be observed. Regarding velocity, there is a significant velocity increase in the section of the bridge, a decrease in the floodplain of the upstream zone, and another increase in the floodplain of the downstream zone.

(5) The methodology developed in this publication is useful for the preliminary identification of the best design option, which meets the requirements of flood hazard regulations, while subsequent studies should focus on other important aspects, with special regard to sediment transport and erosion in piers of bridges.

**Author Contributions:** Conceptualization, J.M.-L. and J.C.-M.; Data curation, V.E.-R.; Formal analysis, J.M.-L., A.G.-G. and V.E.-R.; Investigation, J.M.-L. and J.C.-M.; Methodology, J.M.-L. and J.C.-M.; Software, J.M.-L.; Supervision, J.Á.S.-N.; Validation, J.C.-M. and V.F.-R.; Writing—original draft, J.M.-L.; Writing—review & editing, J.C.-M., A.G.-G. and V.F.-R. All authors have read and agreed to the published version of the manuscript.

**Funding:** This research received no external funding.

**Acknowledgments:** This work has been partially subsidized by the Research Group "Analysis of Continental Sedimentary Basins" of the Government of Aragon and FEDER Funds to which gratitude is transmitted.

**Conflicts of Interest:** The authors declare no conflict of interest.

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
