# Peer review of "Comparative Hydrodynamic Analysis by Using Two−Dimensional Models and Application to a New Bridge"

_water, doi:10.3390/w12040997_

Round 1

Reviewer 1 Report

The paper deals with the comparison of the results of 2D numerical simulations with and without a new infrastructure (a bridge in the proposed application). The aim of the methodology is to assess the impact of the new bridge on the flood hazard in the river and floodplains. The paper confirms the usefulness of 2D numerical models in this kind of applications, and takes advantage of state-of-the-art tools (Iber, SHEE). The application is interesting. However, I think that the Authors overlooked some important aspects in the presentation of their work. For this reason, I believe the paper can be considered for publication only after a major revision.

Major issues:

The abstract is too long, and some ideas are repeated twice (first in general, and then for the application). It should be 200 words maximum, and maybe more focused on the methodology in general (rather than on the results of the application). Introduction (section 1). In my opinion, this is the weakest part of the paper. The Authors should add an overview of previous relevant works in the literature about other existing methodologies to assess the impact of hydraulic structures in rivers, and in particular about the use of numerical models to do it (maybe some works can be found in the context of infrastructure for flood protection). This is intended to provide a literature framework for the proposed methodology and to highlight the novelty of the paper. At the same time, I suggest reducing the number of references to Iber and SHEE software, which are definitely too many and not all related to the topic of this paper. Only the most relevant publications should be cited. Case study (section 2.3). More information should be provided, including: the length of the river stretch considered in the simulations and the size of the study area (do figures 2-5 represent the whole domain or just a part of it?); the number of cells in the mesh; the location of the boundary conditions (XX km upstream/downstream of the bridge). Moreover, it is not clear whether these simulations are performed with a constant inflow discharge (please provide the value) or with a flood wave (unsteady flow) as upstream boundary condition. Another thing that should be specified is whether the bridge piers are included in the mesh (for example as “holes” with reflective boundaries) or neglected (if so, please justify): this can have an impact on local velocities. Results (section 3). I suggest presenting the results slightly differently, separating the section in two sub-sections. The first one should deal with the selection of the best design option: the three options should be described (lines 183-189) and the results should be compared (figure 3 and related comments); the conclusion of this part is that only option 3 meets the requirements. Then, the second sub-section can provide a more in-depth comparison between simulations in the initial state and in the final state with option 3 (figures 2, 4 and 5 and related comments). Conclusions (section 4). I think that this part should summarize the strengths of the methodology, so I would exclude bullet points (5-7), which are case-specific, from here. I suggest pointing out that this method is useful for preliminary identification of the best design option, which meets the requirements of flood hazard regulations, while subsequent studies should focus on other important aspects (especially sediment transport).

Other minor issues:

Define “two-dimensional (2D)” at the first occurrence in the text, and then use the acronym “2D” throughout the paper. Lines 116-118. Are these source terms all considered in the Jalon River simulation? Lines 121-122. Please check the sentence. Moreover, notice that bidirectional (going two ways along a direction) is different from two-dimensional (no predefined direction in the horizontal plane). Line 136. Remove “of 2 m” (this part is the description of the general methodology, which can be extended to any grid size). Line 137. It should be “has been obtained”. Line 139. Maybe “1 m wide vertical strip”. Lines 140-141. Please provide a reference for FHR. Lines 144-146. The sentence is unclear. Lines 147-154. Is this a Spanish law? Please specify. Line 176. Is the SVF algorithm or the Iber simulation run on the GPU? Line 187. What is a box hydraulic? A culvert? Line 188. Is Figure 1 referred to Option 3? Please specify. Figures 2A-4A-5A. Maybe indicate the bridge position with a dashed line for better comparison. Line 228. Remove “the surcharge”.

Author Response

Dear Reviewer,

Please, find attachment.

Kind regards

Reviewer 2 Report

This paper shows a methodology useful to verify the impact that the implantation of different infrastructures around the river currents could produce. The topic addressed by the research is worthy of investigation. The presented case study is more interesting but the structure of the paper is unclear. 

There is a description of the objectives and hypotheses behind the analysis. However, the authors did not discuss how it may benefit a large international audience.  I suggest to consider the publication of the paper, despite some minor revisions. My detailed comments are listed below:

Introduction it is not explained in detail what scientific gaps authors want to address. 

Lines 60—83. I suggest to move this in the section 2.

Lines 58—59 are not in the topic of the paper. I suggest to remove them and to add a reference.

Fig. 1a. I suggest to add the geographic coordinates to the study area. Actually, this figure is unclear.

In section 2  it is necessary to add a detailed description of SHEE model.

I suggest to move the subsection 2.3 in a new section describing the test case. It is not a methodology.

I suggest to split the section 3 in one section describing the obtained results and another section with the relative discussion.

The discussioneshould describe  positive aspects, observed deficiencies, and suggestions on how to improve them. The discussion of results should not be schematic or by points. I suggest to add a rigorous discussion (with comparisons with international studies) in a new section. 

Author Response

(The authors gave the same response as above.)

Reviewer 3 Report

1.Is there any secondary flow occured in the river bend flow ? Some discussions about the curvature of river are suggested in the paper.

2.Besides the water depth and velocity, the section width of the main river channels are needed to be considered in the simulation. In the paper no information of river width is given for the readers. 

3. It is suggested that different flow velocity induced the potential local scour around the bridge should be investigated.

Author Response

(The authors gave the same response as above.)

Round 2

Reviewer 1 Report

The paper has improved from the original version, but I still have some suggestions.

In the following, the line numbering refers to the “clean version”.

  1. Avoid repeated sentences (lines 21-22 and lines 23-24).
  2. I suggest moving lines 54-69 to paragraph 2.1, and lines 70-76 to paragraph 2.3. Only the main reference for each model should be cited in lines 48 and 50. Pay attention that part of the general description of SHEE in the introduction is already repeated in paragraph 2.3 (lines 151-153). In addition, I still feel that most of these references for Iber and SHEE are not relevant for this paper (in particular, conference abstracts should not be cited).
  3. Description of SHEE. Please remove lines 190-196 (the subject is definitely outside the scope of the paper).
  4. The Authors should not anticipate conclusions before showing results (lines 265-266: this sentence should be at the end of the paragraph). Moreover, Figure 2 (and related comments) belongs in paragraph 4.2.
  5. I also think that another Reviewer suggested adding a bit of discussion at the end of paragraph 4, but his/her suggestion has not been accepted. Why?
  6. Carefully check the paper, maybe with the help of an English native speaker. Some sentences are awkward (e.g. line 152-153 “with respect to the considering the entire existing floodplain”) or wrong, e.g. line 121 “made that with” (“that” should not be there), line 123 “assumptions all where…”, line 142 “coverage acquired” (the verb should be passive).

Author Response

Please, find attachment
